# Safe spaces for youth mental health: A scoping review

**Salima Meherali[1]\*, Saba Nisa[1], Yared Asmare Aynalem[1], Adeyinka G. Ishola[2], Zohra Lassi[3]**

1 Faculty of Nursing, University of Alberta, Edmonton, Canada, 2 Department of Nursing, University of Ibadan, Ibadan, Nigeria, 3 School of Public health, University of Adelaide, Adelaide, Australia

\* meherali@ualberta.ca

## Abstract

### Introduction

Mental illness is a significant challenge during adolescence and youth period, posing a threat to individuals' mental health, well-being, and productivity. Despite the global burden, comprehensive evidence on the use of youth-safe spaces to improve their mental health has been limited. Therefore, this review aims to explore the existing literature on the role of safe spaces in shaping the mental health outcomes of youth.

### Methods

We followed the Joanna Briggs Institute (JBI) scoping review guidelines. This review focused on individuals aged 10 to 25. It explores safe spaces for youth, including community centers, schools, clubs, and online forums, and their role in promoting youth mental health. We conducted a comprehensive search using PubMed/MEDLINE, PsycINFO, Web of Science, Scopus, Google Scholar, and grey literature sources. Study selection and screening were done using Covidence software, with two independent reviewers applying predefined criteria. We used the standardized table for data extraction; findings were presented using graphical and tabular formats alongside narrative synthesis. Reporting followed the PRISMA extension for scoping reviews (PRISMA-ScR) framework.

### Results

The review included 23 studies from various regions, notably North America (USA) and Europe. These studies found that safe spaces, primarily within schools, offered youth mental health support, resources, and guidance. Additionally, community organizations, outreach programs, and primary care clinics were identified as safe spaces to enhance the mental well-being of young adults. The interventions used in these safe spaces included cognitive-behavioral therapy, mindfulness programs, and multi-component approaches. Positive outcomes included reduced posttraumatic stress disorders, anxiety, and substance use, along with improved mental well-being and interpersonal relationships. However, there needs to be more focus on methodological diversity and research in other

**Data availability statement:** All relevant data are within the manuscript and its Supporting Information files.

**Funding:** The author(s) received no specific funding for this work.

**Competing interests:** The authors have declared that no competing interests exist.

regions. Geographic imbalances exist, and evidence beyond schools and communities as safe spaces is limited. Intersectional factors are often overlooked.

## Conclusion

This review emphasizes the significant impact of safe spaces on youth mental health. It suggests that fostering supportive environments within schools, recreational clubs, and communities can significantly benefit youth mental well-being. The findings highlight the need to expand safe space initiatives to address young people's challenges during their developmental stage.

## Introduction

Young people (10–25 years) represent a critical stage of human development characterized by changes in physical, cognitive, emotional, and social domains as individuals transition from childhood to adulthood [1,2]. Approximately one-fourth of the world's population falls within this age range [3,4]. Mental illness is a significant challenge to an individual's overall health, well-being, and productivity. Research shows that 50% of mental health issues emerge before the age of 15, and 75% develop by the age of 25 [5,6]. Mental health disorders in youth include a range of conditions such as depression, anxiety disorders, eating disorders, substance use disorders, attention-deficit/hyperactivity disorder (ADHD), post-traumatic stress disorder (PTSD), and schizophrenia, as first-episode psychosis and diagnosis of schizophrenia are common in the 15–25 age group. These disorders can have profound implications for various aspects of development, such as physical and social development. Additionally, untreated mental health issues can hinder academic performance, leading to difficulties in attaining educational and career goals [7–9]. Furthermore, financial independence and autonomy may be compromised as young adults struggle to cope with the debilitating effects of mental illness [10,11]. Neglecting mental health, a critical component of overall well-being, significantly contributes to global morbidity and mortality rates. Youth mental health programs such as mentoring initiatives, collaborative mental health promotion, and FRIENDS for Life initiatives offer diverse treatment programs to assist them in dealing with significant mental health conditions [12,13]. With proper support, adolescents and young adults can better manage their mental health and lead fulfilling lives. However, many young people find that the existing treatment programs do not fully meet their needs or are designed for the adult population [14,15]. The primary strategy for mitigating the adverse mental health outcomes of mental ill-health among young people is to disseminate comprehensive education regarding the nature and progression of mental health issues [16]. Community-wide awareness campaigns, anti-stigma endeavors, and mental health promotion initiatives have succeeded in several countries [13,16,17].

Safe spaces are those places where adolescents and youth feel secure, accepted, and empowered to express themselves authentically without fear of judgment or harm [18,19]. While traditionally associated with schools, community centers, and recreational facilities, safe spaces now extend to online platforms, social media networks, and peer support groups, reflecting contemporary youth's evolving needs and preferences [13,17]. Within these spaces, young people navigate myriad stressors and transitions, ranging from academic pressures and conflicts to identity exploration and societal expectations, which can profoundly impact their mental well-being [20,21]. Recognizing the importance of safe spaces in fostering positive mental health outcomes, this scoping review aims to map the existing literature on the role of safe spaces in shaping the mental health trajectories of youth. We aim to map the diverse

array of research to delineate critical themes, gaps, and areas for future inquiry, guiding future research and policy through a comprehensive review of available evidence.

## Materials and methods

This scoping review is based on the Joanna Briggs Institute (JBI) scoping review guidelines [22]. The research question guiding this scoping review is: *What is the current evidence on the impact of youth safe spaces on mental health outcomes?*

### Eligibility Criteria

This scoping review follows the Population, Concept, and Context (PCC) framework to set/ assess eligibility criteria. It focuses on individuals aged 10 to 25 (**population**). It explores youth safe spaces and supportive environments tailored for this age group, including community centers, school clubs, and online forums (**concept**). The review examines mental health outcomes such as decreased depression, anxiety, and stress, and increased resilience and self-esteem among youth in safe spaces. This study includes studies conducted globally (**context**). We included original research studies published only in English that utilized quantitative, qualitative, or mixed methods designs. We excluded reviews (literature reviews, meta-analyses, and systematic reviews), opinion pieces, editorials, commentaries, and studies unavailable in full-text accessibility.

### Search strategy

After consultation with the team, a comprehensive search strategy was meticulously crafted by a seasoned librarian specializing in reviews (MK) at the University of Alberta. The search strategies included prominent electronic databases such as PubMed/MEDLINE, PsycINFO, Web of Science, Scopus, and Google Scholar, and we also searched for grey literature from 1996 to 5/11/2023. Search terms tailored to each database included Medical Subject Headings (MeSH) terms and keywords related to "youth safe spaces" and "mental health outcomes" to ensure a thorough search of the literature. Truncation, Boolean operators, and phrase searching were also employed to maximize sensitivity and specificity (Supplementary File 1).

### Study selection

The study selection and screening were carried out using Covidence software. Two independent reviewers (YA and SN) completed screening for titles and abstracts against the predefined inclusion and exclusion criteria. Potentially relevant studies progressed to full-text reviews, where the same reviewers independently assessed them for final eligibility. Any discrepancies during this process were resolved through discussion or consultation with a third reviewer (SM). The study selection process was documented using a Preferred Reporting Items for Systematic Reviews and Meta-Analyses Extension for Scoping Reviews (PRISMA-ScR) flow diagram to ensure transparency and rigor [23] (Supplementary File 2). Initially, we identified 8,452 studies from databases and registers. After screening, we removed 4,597 references due to duplicates. We screened 3,855 studies for eligibility and included 23 in the final review (Fig 1).

### Data extraction

We meticulously designed the data extraction table for this study to ensure a comprehensive capture of relevant information from selected studies. The two reviewers (SN and YA) followed JBI guidelines for data extraction using a Word document and resolved discrepancies through discussion or consultation with a third reviewer. We extracted information such as

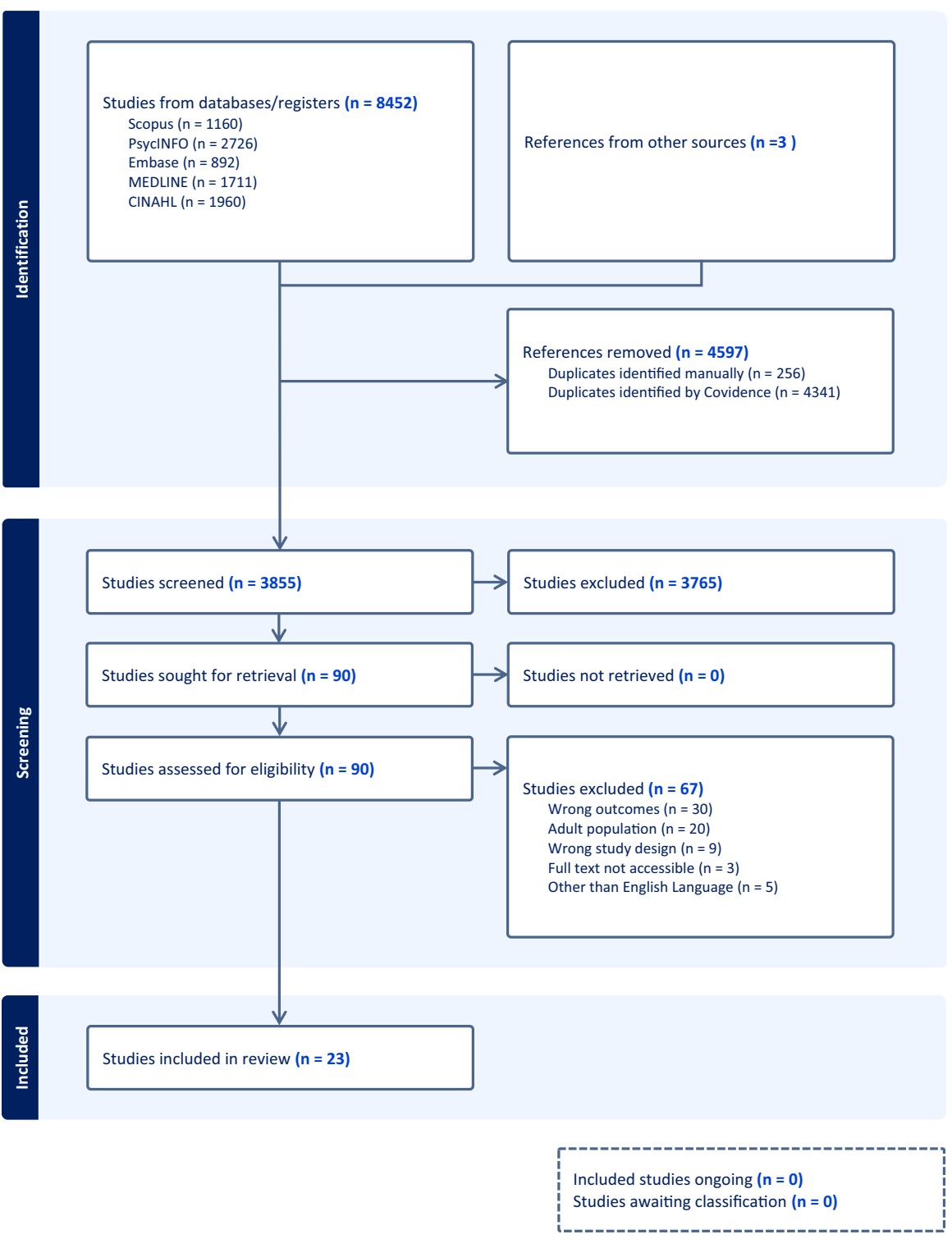

**Fig 1. Prisma Flow Chart.**

**Table 1. Data extraction: Safe spaces for youth mental health.**

| S/NO | Author/year | Country | Study Design | Age | Sample Size | Types of Safe Space | Intervention | Outcome |
|---|---|---|---|---|---|---|---|---|
| 1 | Mora et al., 2022 | USA | Cohort | 15.6 years | n = 537 | School and Community | Extracurricular Activities (unstructured community-based activities) | Reduce PTSD |
| 2 | Fjermestad et al., 202 | Norway | Observational | 11.6 years | n = 82 | School | School-based 'FRIENDS for Life' program 10-session CBT for youth anxiety prevention and treatment. | Reduced anxiety symptoms |
| 3 | June T. Forsberg & Jon-Håkon Schultz, 2022 | Gaza | RCT | 9 and16 years | n = 300 | School | Three-day training BLP-2 implementation for 11 school | Reduced stress symptoms |
| 4 | Curtin et al., 2015 | USA | RCT | 13–18 years | n = 9 | School | Mentoring Program | Enhanced self-esteem. Reduced social anxiety. Improved overall quality of life. |
| 5 | Gigantesco et al., 2015 | Italy | RCT | 14–18 years | n = 234 | School | School-based mental health program | Enhanced psychological well-being. Increased life satisfaction. |
| 6 | Haugland et al., 2020 | Norway | RCT | 14 years | n = 313 | School | Brief and Standard School-Based Cognitive-Behavioral Interventions | Reduced level of Anxiety |
| 7 | Katz et al.,2019 | USA | Cross-sectional | 12–19 years | n = 54 | Outreach | Group Public Speaking Task for Adolescent | Reduction of the level of stress/depression |
| 8 | Roche J et al.,2022 | USA | Effectiveness-implementation hybrid design | 14–18 years | n = 110 | Community-based Primary care clinic | SafERteens-PC program | Reductions in severe peer aggression, anxiety, and substance use consequences |
| 9 | Wasserman D et al.,2015 | European Union countries | Cluster-RCT | Median age 15 years | n = 11,110 | School | Three interventions: Question, Persuade, and Refer (QPR), make the Youth Aware of Mental Health Programme (YAM), and screening by professionals (ProfScreen) with a referral of at-risk pupils. | Significant reduction in incident suicide attempts and severe suicidal ideation compared with the control group. |
| 10 | Kang Y et al.,2018 | USA | RCT | 11.79 years | n = 100 | School | School-based mindfulness meditation program | Improvement in emotional well-being |
| 11 | Rickard et al., 2023 | China | Mixed Method Study | 15.1 years | n = 153 | School | Positive education programme | Reduction in Depression, anxiety, autonomy, gratitude, mindfulness, satisfaction with life, and relatedness levels |
| 12 | Mahmood-abadi et al.,2023 | Iran | Quasi-experimental study | 15 years | n = 30 | Outreach Program | Positive Youth Development (PYD) Program | Reducing aggression behaviours (physical aggression, anger, and hostility) and only affected adolescent verbal aggression. |
| 13 | McAllister et al., 2018 | Australia | Mixed Method Study | 13 years | n = 850 | School | collaborative mental health promotion program | Self-efficacy, resilience, and coping strategies in mental health |
| 14 | Topooco et al.,2018 | UK | RCT | 15–19 years | n = 70 | School and Outreach programs | Chat- and internet-based cognitive–behavioural therapy | Reduced symptoms of depression |
| 15 | Asanjarani, F. & Asgari M. 2021 | Iran | Quasi-experimental study | 14 –16 years | n = 1,008 | School | Social and Mental Empowerment Program (SMEP) | lower difficulties as measured by SDQ, and higher pro-social behaviors. |
| 16 | DeBiase et al., 2021 | USA | Interventional | 16–17 years | n = 5 | School | Multi-component Positive Psychology Intervention (PPI) | Increased happiness, improved classroom behavior, and life satisfaction, |
| 17 | Denny et al., 2019 | New Zealand | Observational Study | 16 > above | n = 15 | School | school-based health service utilization: | use of SBHS was associated with poorer health outcomes, - high levels of depressive symptoms but decreased suicide attempt |

*(Continued)*

**Table 1.** (Continued)

| S/NO | Author/year | Country | Study Design | Age | Sample Size | Types of Safe Space | Intervention | Outcome |
|---|---|---|---|---|---|---|---|---|
| 18 | Anttila et al., 2021 | Finland | Quasi-Experimental Study | 12–17 years | (n = 201 in the intervention school, n = 157 in the control school) | School | The IT-based program "DepisNet," a user-friendly and feasible support system developed for adolescents with depression. in this study | There were no statistically significant differences between the groups in any outcomes (depression, quality of life, self-esteem, self-efficacy). Regarding adolescents' quality of life, the observed change was more positive in the intervention group |
| 19 | Chang et al., 2018 | Taiwan | Quantitative | 17.14 years, mean age | n = 84 | School | E-course program on stress management, refusal skills, pros of drug use, | There was a significant group time interaction concerning stress management, refusal skills, pros of drug use, and drug use resistance self-efficacy, excluding cons of drug use |
| 20 | Ponsoda et al., 2017 | Spain | Interventional | 16 to 18 years | n = 30 | School | out-of-school mindfulness program for stress reduction and emotional well-being | Reductions in perceived stress increase in levels of optimism and five specific mindfulness skills. |
| 21 | Curtin et al., 2016 | USA | Pre-post design | 13 to 18 years | n = 9 | School | Expanding Horizons: A Pilot Mentoring Program Linking College/Graduate Students and Teens With ASD | improvement in self-esteem, social anxiety, and quality of life |
| 22 | Eslami et al., 2023 | Iran | RCT | 13 to 18 years | n = 106 | School | school-based social skills training (SSTS) educational program to prevent adolescents' problem behaviours | Lower levels of MPBI result in a significant difference between groups founded on SST. This suggests that SST was effective in improving social competence and preventing problem behaviours among male adolescents. |
| 23 | Connolly et al, 2022 | USA | Mixed-methods design | 12 to 17 years | n = 22 | School | Hope 4 Boys is a youth program that aims to reduce recidivism and prevent male youth from interacting with the Department of Juvenile Justice. | Significant increases in hope, life satisfaction, and resilience scores |

authors, publication year, country, study design, participants' details, types of safe spaces and interventions, mental health outcomes assessed, and study methods pertinent to the review question(s). Missing data were documented, and study authors were contacted when feasible. Analysis was based on available information, with limitations noted. (Table 1).

## Data synthesis

The findings of this scoping review are presented in graphical and tabular formats. Additionally, we utilized narrative synthesis to further elaborate on qualitative conclusions and explain how the results relate to and address the research objectives. We analyzed quantitative data using descriptive numerical summaries and qualitative content analysis techniques.

## Findings

We organized our findings based on the following categories.

### Features of the included studies

Among the 23 included studies, most predominantly adopt quantitative methodologies, with experimental designs being the most frequently reported (14 studies) [24–37]. Other observational quantitative approaches, such as cohort [38–40] and cross-sectional studies [41–43], are

also represented. In addition, our review includes three mixed-method studies, demonstrating a combination of quantitative and qualitative analysis [42,44,45]. We found that most of the studies included population age ranging from 9 to 19 years, with the most common age group falling between 14 to 16 years, comprising a maximum sample size of 11,110 in quantitative studies [24].

Regarding geographical distribution, the evidence from our included studies reveals diversity across continents. North America (USA) emerges as the most represented region, with eight studies providing safe spaces for youth to improve their mental health and resilience [26,29,30,34,38,41,45,46], followed by Europe with seven studies, including countries such as Norway [24,28,39], Italy [27], the UK [32], Finland [35], and Spain [36]. Asia contributes by five studies, represented by China [44], Iran [31,33,37], and Taiwan [40], while Oceania, including Australia [42] and New Zealand [43], contributes by two studies. This distribution underscores the global scope of our research findings.

## Types of safe spaces

Our review found that most of the existing literature on safe spaces for youth includes schools as the most popular in providing supportive environments, access to resources, guidance, and opportunities for engagement in meaningful activities (18 studies). These studies identify different safe spaces within school settings, such as school-based health education, covering topics such as sexual health, mental health, substance abuse prevention, and healthy lifestyle choices in a confidential and non-judgmental setting [24–28,30,32–34,36–40,42–45]. Moreover, other than schools, community organizations and youth outreach programs are highly effective and provide safe spaces for youth to meet their mental health needs [31,32,38,41]. However, some safe spaces are described as encompassing both school and community environments, while others are associated with outreach programs connected to schools [32,45]. One study identifies a community-based primary care clinic as a safe space. Furthermore, our findings suggest some community-based primary clinics that improve youth mental health [29]. The effectiveness of these programs strongly emphasizes creating safe environments within educational settings, with specific initiatives extending beyond schools to involve community engagement.

## Safe space interventions

The included studies identify various intervention types implemented in safe spaces such as schools, communities, and outreach groups to improve mental health outcomes. In schools, initiatives like the 'FRIENDS for Life' program [39], which focuses on building resilience and promoting social and emotional skills, aim to foster friendships and strengthen social connections among individuals. Our findings also revealed that leveraging technology for internet-based therapy or peer support networks can improve accessibility and engagement among youth and adolescents [47,48]. Moreover, a three-day training session on Building Learning Power (BLP-2), a strategy to promote resilience, growth mindset, and self-awareness and foster mental well-being through innovative educational approaches in schools, is notable [25]. Additionally, various mental health programs, including cognitive-behavioral therapy (CBT) interventions and mindfulness meditation programs, are integrated into school settings [27,28,36,44], emphasizing positive education and social skills training. Community-based interventions extend beyond school grounds, encompassing activities like extracurricular programs [37,38], mentoring initiatives [26], and collaborative mental health promotion efforts [30,42]. Notably, one intervention connects college/graduate students with teens diagnosed with ASD, aiming to provide support and expand horizons [26,33].

Other interventions used include Multi-component interventions that utilize strategies such as Question, Persuade, and Refer (QPR) and Youth Aware of Mental Health Programme (YAM), complemented by professional screening and referral systems. These interventions comprehensively address mental health needs by combining various evidence-based strategies and resources [24,31,34]. Additionally, interventions used by community safe spaces, such as group public speaking tasks and the SafERteens-PC program, are implemented to address mental health concerns [24,29,40,41,43]. The SafERteens-PC program is a youth-focused initiative to promote safety, well-being, and mental health among teenagers. It involves education and resources aimed at enhancing teen awareness of safety issues, such as substance abuse prevention, violence prevention, and mental health promotion.

## Mental health outcomes

The safe space and intervention programs demonstrated effectiveness in producing a range of outcomes across various domains, notably enhancing mental health and overall well-being. Mental health and well-being outcomes include reductions in PTSD [38], anxiety [24–26,28,29,32,36,39,41,44], gambling symptoms, and a decrease in intrusive thoughts, such as those related to obsessive-compulsive disorder (OCD) or suicidal ideation [24,40,43]. Behavioral and functional outcomes involve reductions in aggressive behaviors such as physical aggression, anger, and hostility, along with enhancements in social competence and problem behaviors [24,29,31,40–42]. Interventions also reduce substance use consequences, significantly decrease suicide attempts and severe suicidal ideation, and improve classroom behavior [24,43].

Psychological and cognitive realms have witnessed positive change, including boosts in self-esteem [26,37], reductions in social anxiety [26,44], and enhancements in quality of life and psychological well-being among youth [26,36,40]. Moreover, there has been an increase in life satisfaction [27,44,45] and happiness, accompanied by improvements in self-efficacy, resilience, coping mechanisms, stress management, and resistance to drug use[24,29,34,42,45]. Interpersonal and social dimensions have also seen remarkable improvements, with interventions fostering better interpersonal relationships, reducing personal stigma toward mental health conditions, and promoting pro-social behaviors [26,27,34,36,40,44]. Interventions improved mental health outcomes for youth and focused on improving staff well-being and morale and increasing public satisfaction with mental health services at organizational and systemic levels [30,33].

## Discussion

This review mapped evidence on how safe space interventions can enhance mental health and well-being among youth aged 10–25. Our study found existing evidence with a predominant quantitative methodology, like experimental designs, indicating rigorous data collection and analysis. While qualitative approaches received less attention, incorporating mixed-method studies showed a commitment to triangulating findings and comprehensively capturing the complexity of teenage experiences. However, it is essential to integrate qualitative methodologies, which have yet to receive less attention in the literature. These approaches, including interviews, focus groups, and participant observation, offer valuable insights into youths' subjective experiences and perspectives within safe spaces [47–49].

The review discusses the available safe spaces for supporting youth mental health and well-being. Schools are highlighted as crucial environments for youth development and support, offering structured programs and interventions to address mental health issues. This result aligns with other research emphasizing the role of educational settings in providing a

supportive environment for mental health interventions [49,50]. Safe spaces within schools go beyond physical environments to include supportive relationships, accessible resources, and inclusive policies [51]. However, we observed a shortage of research focusing on community centers and health clinics providing safe environments to young people (3 studies only). Limited safe spaces might contribute to social isolation, a lack of support networks, and difficulties in accessing mental health resources and services among youth [52,53]. Moreover, it highlights the need to broaden safe space interventions to include young individuals who may not have access to school-based programs. This finding aligns with the recommendation to design programs and places conducive to mental health for youth [54,55].

The findings of our review are congruent with previous studies that documented interaction-based interventions in schools and communities as effective in improving youth mental health outcomes, including reducing symptoms of anxiety, depression, and PTSD. These interventions also enhance social competence, interpersonal relationships, and behavioral outcomes, indicating a broader impact on youth development and functioning [49,56,57]. The review also suggests the potential of digital approaches like mobile apps and social media platforms to enhance the accessibility and engagement of safe space interventions, mirroring studies on digital mental health interventions [49,50]. Our findings also revealed that leveraging technology for internet-based therapy or peer support networks can improve accessibility and engagement among young adults. This emphasizes a holistic approach to mental health promotion by targeting various aspects of youth well-being across diverse settings. Multi-level interventions incorporating components like economic empowerment, peer support, and cognitive behavioral therapy have also proven effective in improving mental health among vulnerable youth; this is concurrent with previous studies on multi-level interventions on mental health outcomes among youth in sub-Saharan Africa [58,59]. A holistic understanding of these outcomes highlights the transformative potential of safe space interventions [60]. Disparities in safe space provision globally reveal unequal access to mental health support for youth. While North America and Europe lead, Asia and Oceania have fewer studies, indicating a need for more research and interventions, especially in low- and middle-income countries with limited resources promoting overall youth well-being.

This scoping review has several strengths and limitations. A key strength is its inclusion of studies from around the world, offering a comprehensive global perspective. The review also followed a systematic and transparent methodology, enhancing the reliability of the findings. However, notable limitations include the exclusion of non-English studies, which may have overlooked relevant research. Additionally, the potential for publication bias and the heterogeneity in study design, sample size, and quality could affect the generalizability and comparability of the results. Furthermore, the scoping methodology focuses on summarizing available evidence rather than critically evaluating or scoring the quality of the studies.

## Implications

The review findings offer valuable insights that can guide researchers and practitioners in shaping their approach to youth mental health interventions and safe spaces. Several key gaps were identified, which can inform future research and practice.First, most studies were from high-income regions (e.g., North America and Europe), with limited representation from low- and middle-income countries, particularly in Asia and Oceania. This highlights the need for more research in underrepresented regions to assess the effectiveness of safe spaces for mental health and address the unique challenges faced by youth in these areas. Funding agencies should prioritize research in these regions to fill existing knowledge gaps.Second, although the study focuses on ages 10–25, most interventions primarily target younger adolescents, neglecting the needs of older youth. Future interventions should address the mental health needs of

older youth to ensure inclusivity across the entire age range.Third, intersectional factors such as gender, ethnicity, and socioeconomic status remain underexplored. Future studies should prioritize these factors to better understand how different identities influence the effectiveness of safe spaces and interventions.

While the review discusses implications for policy and practice, specific actionable recommendations are limited. We suggest that future research incorporate methodological diversity to capture a comprehensive view of the field. The broad age range and significant sample size underscore the importance of inclusive research designs, ensuring that diverse youth experiences are represented. Additionally, the scarcity of safe spaces outside schools highlights the vulnerability of youth to mental health challenges, emphasizing the need to expand interventions beyond educational settings. Policy development should focus on creating safe spaces in community centers and other relevant environments, with a clear focus on resource allocation and intervention guidelines.Practitioners, including mental health professionals, educators, and community workers, would benefit from strategies tailored to the specific mental health needs of youth. Furthermore, addressing underexplored areas, such as cultural sensitivity in interventions and gender-specific mental health needs, is crucial. Ensuring the sustainability of safe space initiatives through ongoing funding and support is vital for creating long-term impacts.Future research could explore innovative digital approaches, such as using social media platforms and virtual safe spaces, to enhance youth mental health while addressing concerns around digital abuse. Longitudinal studies are essential for assessing the lasting effects of safe space interventions on youth mental health and well-being. Additionally, funding frameworks should be designed to ensure continuous support for these initiatives, prioritizing long-term investment to sustain their impact and effectiveness.

## Conclusion

Safe spaces play a crucial role in addressing the mental health challenges young people face by providing a supportive environment where they can seek accurate information and assistance and develop healthy behaviors. While safe spaces are instrumental, further research is necessary to refine and evaluate interventions within these environments. While schools are commonly recognized as safe spaces, broadening interventions to various settings is essential to addressing a broader range of outcomes and meeting diverse needs of young adults globally. Safe space interventions promote mental health and well-being among youth globally. Geographical disparities underline the need for global collaboration in fostering safe spaces for youth to improve mental health outcomes effectively.

## Supporting information

**Supplementary File 1. Supplementary files on database searching Strategies.**
(DOCX)

**Supplementary File 2. Preferred Reporting Items for Systematic Reviews and Meta-Analyses extension for Scoping Reviews (PRISMA-ScR) Checklist.**
(DOCX)

## Author contributions

**Conceptualization:** Salima Meherali, Zohra Lassi.

**Data curation:** Saba Nisa, Yared Asmare Aynalem, Adeyinka G. Ishola.

**Formal analysis:** Saba Nisa, Yared Asmare Aynalem, Adeyinka G. Ishola.

**Methodology:** Salima Meherali, Saba Nisa, Yared Asmare Aynalem, Adeyinka G. Ishola.

**Supervision:** Salima Meherali, Zohra Lassi.

**Validation:** Saba Nisa.

**Visualization:** Salima Meherali, Saba Nisa, Yared Asmare Aynalem.

**Writing – original draft:** Salima Meherali, Saba Nisa, Yared Asmare Aynalem, Adeyinka G. Ishola.

**Writing – review & editing:** Salima Meherali, Saba Nisa, Yared Asmare Aynalem, Zohra Lassi.

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
