## [Decision Letter · Decision Letter 0]

17 Dec 2024

PONE-D-24-28524Safe Spaces for Youth Mental Health: A Scoping ReviewPLOS ONE

Dear Dr. Nisa,

Thank you for submitting your manuscript to PLOS ONE. After careful consideration, we feel that it has merit but does not fully meet PLOS ONE’s publication criteria as it currently stands. Therefore, we invite you to submit a revised version of the manuscript that addresses the points raised during the review process.

The authors have presented a systematic review and the writing does bring forth the key concepts and the characteristics of the studies in the given domain of safe spaces and the interventions used in such spaces with some evidence of effect on mental health outcomes. While a detailed analysis is out of scope for this review, even for a scoping review the authors should attempt to classify in the introduction as well as in the abstract as what they classified as safe spaces. This is more clear from looking at the search terms in the supplement but it would benefit the reader to know from abstract itself as to what is the main domain of this review.  The reviewers have pointed out some language errors which the authors should address. The authors have highlighted some knowledge gaps specific to how the research is less in some countries in this domain and how longitudinal research may add further. Yet these are more general limitations present in most of the literature considering a larger number of studies are cross-sectional as  well as this domain is not so well researched. It would serve better to expand on how the studies could have improved and what are the barriers that could help in more research in this area, such as whether it is more of an awareness issue related to safe space, or there is less attention given to mental health. Expanding on the knowledge gaps would make the review more meaningful. 

We look forward to receiving your revised manuscript.

Kind regards,

Aditya Pawar

Guest Editor

PLOS ONE

2. As required by our policy on Data Availability, please ensure your manuscript or supplementary information includes the following:

Reviewers' comments:

Reviewer's Responses to Questions

**Comments to the Author**

1. Is the manuscript technically sound, and do the data support the conclusions?

Reviewer #1: Yes

Reviewer #2: Partly

Reviewer #3: Yes

Reviewer #4: Yes

2. Has the statistical analysis been performed appropriately and rigorously? 

Reviewer #1: I Don't Know

Reviewer #2: N/A

Reviewer #3: Yes

Reviewer #4: I Don't Know

3. Have the authors made all data underlying the findings in their manuscript fully available?

Reviewer #1: Yes

Reviewer #2: Yes

Reviewer #3: Yes

Reviewer #4: Yes

4. Is the manuscript presented in an intelligible fashion and written in standard English?

Reviewer #1: Yes

Reviewer #2: Yes

Reviewer #3: Yes

Reviewer #4: Yes

5. Review Comments to the Author

Reviewer #1: The manuscript is well written in a lucid style and highlights an important aspect of emotional ,social and holistic development of youths and young adults. More research and actions need to be undertaken in the community development of safe spaces.

Reviewer #2: Thank you for the opportunity to review this paper. This manuscript tackles an important topic by exploring the role of safe spaces in supporting youth mental health. It addresses a critical area of research with global significance. Here are my comments and suggestions:

Abstract: Under method section - “Followed”, make it lower case to “followed”

- Introduction: Make it consistent “mental health” where possible. I see to many different terminology used for the same.

- “Mental health disorders in youth include a range of conditions such as depression, anxiety disorders, eating disorders, substance use disorders, attention-deficit/hyperactivity disorder (ADHD), and post traumatic stress disorder (PTSD)”. Consider adding schizophrenia, as first episode psychosis and diagnosis of Schizophrenia tends to be very common in age group between 15-25 years old.

- Search Strategy : Authors indicated that search for Grey literatures was done. Explain if those literatures were used for the review or excluded. If used then I recommend to indicate separately in the PRISMA flow chart.

- Data Extraction : Please include details on how you handled missing data in the included studies during extraction.

- Results and discussion :You mentioned the imbalance in geographical distribution earlier in the paper. Consider providing a comparative analysis of interventions across regions to offer deeper insights into the influence of regional differences on intervention efficacy.

- Consider adding recommendations for funding frameworks and longitudinal study designs to ensure the sustainability and long-term impact of these interventions.

- The manuscript effectively highlights schools as key environments for youth mental health interventions but only briefly mentions community-based interventions. Consider addressing how community settings could fill gaps left by school-based programs, such as supporting out-of-school youth.

Reviewer #3: Overall, this is a well-written article. It provides a good case for the need to explore safe spaces and its overall benefit to youth.

Introduction: This section includes a good review of research already present on the topic, along with defining safe spaces and the myriad of environments they can be present. Some considerations can be made with the wording. For example, for the following sentence on page 3: “Mental ill-health is a significant challenge, posing a threat to individuals’ health, well-being, …” It appears redundant to mention mental ill-health and then an individuals’ health again in the same sentence. Also, what is the reasoning behind mentioning “mental ill-health” vs. calling it “mental illness”?

Further along in the same paragraph, it mentions: “Youth mental health programs such as mentoring initiatives, collaborative mental health promotion, and friends for life initiatives…” It seems that the “friends for life” initiative should be capitalized as it seems to be naming a specific initiative.

Materials and Methods:

Authors have done a good job of providing a thorough explanation of the methods used in the review. They can consider a table to synthesize and categorize the final set of studies chosen for the review. For example, how many were initially reviewed, how many were selected and of those how many were school-based, community-based, etc. Providing it in a table or figure format would make visualization easier for the reader.

Under eligibility criteria on page 4, it mentions “The review examines mental health outcomes like decreased depression, anxiety, stress, resilience, and self-esteem among youth in safe spaces.” Authors can consider omitting the word “decreased” or adding the word “increased” before “resilience, and self-esteem” for consistency.

On page 6, under “features of the included studies” section, it states: “North America emerges as the most represented region…followed by Europe…including countries like Norway…”. The word “like” is not necessary, authors can simply list the countries that were included without the word “like”.

On page 7, where “FRIENDS for Life initiative” is mentioned, it would be helpful to provide brief information about what that entails.

Mental Health Outcomes:

It is to note that the body of the review does not have a clearly labelled “Results” section. Authors can consider adding this.

On page 8, it mentions “Mental health and well-being outcomes include reductions in PTSD, anxiety, gambling symptoms, and a decrease in intrusive thoughts”. Please clarify what is meant by intrusive thoughts – was the study looking at OCD, or intrusive thoughts of a suicidal nature, etc.?

Although it is a scoping review, authors can consider mentioning some specific outcomes, such as the range of decrease in depressive symptoms across various studies (e.g. 10-50%), or the range of decrease in anxiety or suicidal thoughts, etc. It would also be interesting to note any differences in outcomes between safe spaces in schools vs. community centers vs. primary care centers, etc.

Discussion:

This section provided a good review of the results, its applicability to the public, and future considerations for research. On page 10, it mentions: “Nevertheless, with valuable insights into mental health intervention within safe spaces, our stay may have limitations”. This appears to be a typo, I believe it meant to say “..our study has some limitations”.

Overall, the paper did a good job at highlighting areas of growth in the community for improving mental health in youth. Authors can consider the points discussed above.

Reviewer #4: The scoping review titled "Safe Spaces for Youth Mental Health" provides a valuable synthesis of the literature concerning safe spaces and their role in enhancing youth mental health. While the limitations have been discussed, there are some additional limitations that need to be addressed or mentioned in the limitations:

1. The majority of included studies originate from high-income regions (e.g., North America and Europe), with limited representation from low- and middle-income countries.

2. Although the study targets ages 10-25, the majority of interventions cater to younger adolescents, neglecting older youth.

3. Intersectional factors such as gender, ethnicity, and socioeconomic status are underexplored.

4. While the study emphasizes implications for policy and practice, specific actionable recommendations are limited.

It would be helpful to provide detailed guidelines for policymakers and practitioners, such as frameworks for implementing safe spaces in diverse contexts.

6. PLOS authors have the option to publish the peer review history of their article (what does this mean? ). If published, this will include your full peer review and any attached files.

**Do you want your identity to be public for this peer review?** For information about this choice, including consent withdrawal, please see our Privacy Policy .

Reviewer #1: No

Reviewer #2: **Yes: ** Mohsin Raza

Reviewer #3: No

Reviewer #4: No

---

## [Author Response · Author response to Decision Letter 1]

23 Jan 2025

Dear editor, Thank you for your feedback and the opportunity to revise our manuscript. We addressed all points raised, revising the abstract and introduction to clearly define "safe spaces," correcting language errors, and expanding the discussion on knowledge gaps by detailing study improvements and barriers to research. We have uploaded a rebuttal letter, a revised manuscript with tracked changes, and a clean version without tracked changes. Thank you for your guidance, and we look forward to your response.

---

## [Decision Letter · Decision Letter 1]

27 Feb 2025

PONE-D-24-28524R1Safe Spaces for Youth Mental Health: A Scoping ReviewPLOS ONE

Dear Dr. Nisa,

Thank you for submitting your manuscript to PLOS ONE. After careful consideration, we feel that it has merit but does not fully meet PLOS ONE’s publication criteria as it currently stands. Therefore, we invite you to submit a revised version of the manuscript that addresses the points raised during the review process.

**I appreciate your making the required edits, the manuscript seems to be in good shape. Kindly incorporate few minor changes as suggested by reviewers. I could see the PRISMA flowchart attached so no addition is needed in that regard. Please make sure that the figures and tables are referenced correctly in the manuscript.**

 Please submit your revised manuscript by Apr 13 2025 11:59PM. If you will need more time than this to complete your revisions, please reply to this message or contact the journal office at plosone@plos.org . Please include the following items when submitting your revised manuscript:

We look forward to receiving your revised manuscript.

Kind regards,

Aditya Pawar

Guest Editor

PLOS ONE

**Journal Requirements:**

Reviewers' comments:

Reviewer's Responses to Questions

**Comments to the Author**

1. If the authors have adequately addressed your comments raised in a previous round of review and you feel that this manuscript is now acceptable for publication, you may indicate that here to bypass the “Comments to the Author” section, enter your conflict of interest statement in the “Confidential to Editor” section, and submit your "Accept" recommendation.

Reviewer #2: All comments have been addressed

Reviewer #3: (No Response)

Reviewer #4: All comments have been addressed

2. Is the manuscript technically sound, and do the data support the conclusions?

Reviewer #2: Yes

Reviewer #3: Yes

Reviewer #4: Yes

3. Has the statistical analysis been performed appropriately and rigorously? 

Reviewer #2: I Don't Know

Reviewer #3: I Don't Know

Reviewer #4: I Don't Know

4. Have the authors made all data underlying the findings in their manuscript fully available?

Reviewer #2: Yes

Reviewer #3: Yes

Reviewer #4: Yes

5. Is the manuscript presented in an intelligible fashion and written in standard English?

Reviewer #2: Yes

Reviewer #3: Yes

Reviewer #4: Yes

6. Review Comments to the Author

**Reviewer #2:**  Thank you for addressing and considering my recommendations. I have no additional comments. I believe this manuscript can be accepted in its current form. I am still not able to see a Preferred Reporting Items for Systematic reviews and Meta-Analyses (PRISMA) flow chart. Adding a PRISMA can significantly enhance the credibility and rigor of the study and helps in reproducibility as well.

All the best.

**Reviewer #3: ** Authors have done a good job with the revision of the paper. Authors can consider the following edits:

1. On page 3, it states “Mental health is a significant challenge, posing a threat to individuals’ health, well-being…”. Authors can consider changing this to “Mental illness is a significant challenge to an individual’s overall health, well-being, and productivity.” Reasoning would be that it is mental illness, not mental health, that is the challenge. Also, mentioning the word “health” twice in the same sentence appears redundant.

2. On page 7, please clarify if “FRIENDS for Life” should be capitalized as such? In the introduction section earlier, it is capitalized as “Friends for Life”. Please make it is consistent throughout the article.

3. The rest of this writer’s previous concerns have been addressed.

**Reviewer #4: ** (No Response)

7. PLOS authors have the option to publish the peer review history of their article (what does this mean? ). If published, this will include your full peer review and any attached files.

**Do you want your identity to be public for this peer review?** For information about this choice, including consent withdrawal, please see our Privacy Policy .

Reviewer #2: **Yes: ** Mohsin Raza

Reviewer #3: No

Reviewer #4: **Yes: ** Nikhil Tondehal

---

## [Author Response · Author response to Decision Letter 2]

27 Feb 2025

We have made the required changes and carefully addressed the points you raised in your feedback. We appreciate your time and thoughtful suggestions, which have helped us improve the manuscript.

---

## [Editor Report · Decision Letter 2]

2 Mar 2025

Safe Spaces for Youth Mental Health: A Scoping Review

PONE-D-24-28524R2

Dear Dr. Nisa,

We’re pleased to inform you that your manuscript has been judged scientifically suitable for publication and will be formally accepted for publication once it meets all outstanding technical requirements.

Kind regards,

Aditya Pawar

Guest Editor

PLOS ONE
---

## [Editor Report · Acceptance letter]

PONE-D-24-28524R2

PLOS ONE

Dear Dr. Nisa,

I'm pleased to inform you that your manuscript has been deemed suitable for publication in PLOS ONE. Congratulations! Your manuscript is now being handed over to our production team.

Kind regards,

on behalf of

Dr. Aditya Pawar

Guest Editor

PLOS ONE